# Experimental Study on Seismic Behavior of Steel Frames with Infilled Recycled Aggregate Concrete Shear Walls

**Lijian Sun [1,2], Hongchao Guo [1,2,*] and Yunhe Liu [1,2]**

[1] State Key Laboratory of Eco-hydraulics in Northwest Arid Region, Xi'an University of Technology No.5 Jinhua Road, Xi'an 710048, China; sunlijian0414@163.com (L.S.); liuyunhe1968@163.com (Y.L.)

[2] School of Civil Engineering and Architecture, Xi'an University of Technology No.5 Jinhua Road, Xi'an 710048, China

\* Correspondence: ghc_1209@163.com; Tel.: +86-132-2700-9454

**Abstract:** Experiments were performed on four specimens of steel frames with infilled recycled aggregate concrete shear walls (SFIRACSWs), one specimen of infilled ordinary concrete wall, and one pure-steel frame were conducted under horizontal low cyclic loading. The influence of the composite forms of steel frames and RACSWs (namely, infilled cast-in-place and infilled prefabricated) on the failure modes, transfer mechanisms of lateral force, bearing capacity, and ductility of SFIRACSWs is discussed, and the concrete type and connecting stiffness of beam–column joints (BCJs) are also considered. Test results showed that infilled RACSWs can increase the bearing capacity and lateral stiffness of SFIRACSWs. The connecting stiffness of BCJs slightly influences the seismic behavior of SFIRACSWs. In the infilled cast-in-place RACSWs, the wall cracks mainly extended along the diagonal direction. The bearing capacity was 2.4 times higher than in the pure steel frame, the initial stiffness was 4.3 times higher, and the displacement ductility factors were 2.44–2.69 times higher. In the infilled prefabricated RACSWs, the wall cracks mainly extended along the connection between the embedded T-shape connectors and walls before finally connecting along the horizontal direction. Moreover, shear failure occurred in the specimens. The bearing capacity was 1.44 times higher than that of the pure steel frame, the initial stiffness was 2.8 times higher, and the displacement ductility factors were 3.32–3.40 times higher. The degradation coefficients of the bearing capacity were more than 0.85, indicating that the specimens demonstrated a high safety reserve.

**Keywords:** steel frame; infilled shear walls; recycled aggregate concrete; semi-rigid connection; seismic behavior

---

**Highlights**

- The practicability of recycled aggregate concrete shear walls (RACSWs) as lateral resistance components of steel structures is investigated.
- An experiment on steel frames with infilled cast-in-place and prefabricated RACSWs (SFIRACSWs) was conducted.
- The effects of concrete type, composite forms of steel frames and RACSWs, and connecting stiffness of beam–column joints are considered.
- The failure modes and transfer mechanisms of lateral force of SFIRACSWs are clarified.
- The main seismic performance indexes of SFIRACSWs are compared with those of pure steel frames.

## 1. Introduction

Recycled aggregate concrete (RAC) can fundamentally solve concrete waste problems not only by reducing environmental pollution from waste concrete, but also preserving natural aggregates and reducing the consumption of natural resources and energy. RAC is one of the main approaches to developing a circular economy and promoting environmentally friendly buildings. The suitability of waste concrete as recycled aggregate for construction projects has been investigated. Puthussery et al. [1] reported that recycled aggregate can be used as a building material for road construction, mass concrete engineering, and lightly reinforced sections, thereby providing ideas for recycling concrete waste.

With regard to the mechanical properties of RAC, the compressive and tensile strengths of recycled coarse aggregate concrete with different sources and strength grades were studied. Tabsh et al. [2], Koenders et al. [3], and Silva et al. [4] found that the strength reduction of RAC is clearer with low-strength coarse aggregate than with high-strength aggregate, and the compressive and tensile strengths of RAC made of 50 MPa coarse aggregate are equal to those of natural aggregate concrete. Bairagi et al. [5] and Oikonomou [6] proposed stress–strain curves of RAC with different aggregate replacement rates in which the constitutive relation of RAC with different aggregate replacement rates was similar, and only the decline stage was different. Ying et al. [7] and Wang et al. [8] studied the diversity of chloride ion diffusion in RAC and analyzed the influence of carbonation modification on the interface properties of RAC. Carbonation can improve the interface properties of RAC, especially when the water–cement ratios of new and old cement mortars are high; thus, the improvement effect is noticeable.

With respect to the components of RAC, Arezoumandi et al. [9] and Choi et al. [10] tested the shear strength of RAC beams under short- and long-term loads and discussed the applicability of the design code, modifying compression field theory to the shear strength of the RAC beams. The axial compressive performance of RAC columns was also investigated. Choi et al. [11] and Xiao et al. [12] found that the maximum axial compressive bearing capacity of RAC columns decreases slightly with an increase in the replacement rate of recycled coarse aggregate, and RAC columns can be used for the load-bearing member of the structure. Wu et al. [13,14] proposed a concept for recycling mixed components based on the recycling technology of large-scale waste concrete blocks and systematically studied thin-walled steel tubular columns, U-shape steel beams, and thin steel-plate walls filled with RAC. By testing RAC-filled square steel tube (RACFST) columns and RACFST columns strengthened by carbon-fiber-reinforced polymer, Chen et al. [15] and Dong et al. [16] found that the RACFST columns exhibited good seismic performance under low axial compression; the aggregate replacement rate demonstrated a minimal influence on the RACFST columns. Fathifazl et al. [17] and Ma et al. [18] studied steel-reinforced RAC (SRRAC) beams and columns and investigated the effect of the replacement rates of recycled coarse aggregate, axial compression ratios, and stirrup ratios on the seismic performance of the SRRAC columns.

For the RAC structures, the seismic performance of two connections was compared and analyzed by testing the RAC beam–column joints (BCJs) [19,20]. Xiao et al. [21] and Wang et al. [22] performed the shaking table test on RAC frames and reported the dynamic response of RAC frames. Tests on RAC shear walls (RACSWs) have been conducted. Peng et al. [23] and Ma et al. [24] found that the existing formulas cannot predict the peak load and failure modes of squat RACSWs and proposed a mixed flexural and diagonal compression mechanism. The dynamic responses and failure modes of ordinary concrete and RAC specimens were compared and discussed by using the shaking table test on RAC frame-shear walls [25].

In general, recycled coarse aggregate is different from natural gallet and pebble aggregate, and its porosity is high, thereby resulting in high water absorption, large dry shrinkage and creep, and poor bond performance of the RAC. Considerable research on the mechanical properties, components, and structures of RAC have demonstrated that RAC shows similar mechanical properties to ordinary concrete and can be widely used in construction after rational design. The main load-bearing

components of RAC structures include recycled coarse aggregates, which were popularized during the construction of towns after the Wenchuan earthquake in 2008.

Steel structure residences have the advantages of strong seismic resistance, high industrialization, recyclability, and reduced resource consumption and construction waste discharge. They are also among the residential structure systems preferred by developed countries at present. In addition to using steel to build steel structure residences, the material development and application technology of enclosure systems have also been assessed. This scenario is a technical problem for new building energy-saving materials and systems, and a social and economic development problem for factory construction of housing in terms of changing construction modes. Tong et al. [26] and Sun et al. [27] tested semi-rigid steel frame-filled concrete shear walls and determined that the energy dissipation of this composite structure mainly depends on the aggregate friction and occlusion between the cracks of filler walls and the yield of shear studs. The structure has multiple transmission paths of horizontal loads and a high safety reserve. Kurata et al. [28] and Guo et al. [29,30] conducted research on a steel-plate shear wall with a semi-rigid steel frame and found that the structure exhibits the advantages of semi-rigid joints with good rotation capability as well as energy absorption through the yield deformation of the steel plate. The hysteretic performance is stable and characterized by the simplified construction and efficient utilization of materials. Wu et al. [31] conducted an experimental study on steel frames with replaceable reinforced concrete walls. Other tests, such as tests on steel frames with fabricated autoclaved lightweight concrete panels, composite lightweight walls, and light-gauge steel stud walls, were conducted under horizontal low cyclic loading [32–34].

Prefabricated construction systems have the advantages of fast construction, stable quality, and energy saving, along with being an environmentally friendly and sustainable development technology. In this study, two aspects of improvements were considered on the basis of the structure of steel frames filled with concrete shear walls as proposed by the previous scholars. On the one hand, the RAC concrete was used to replace the ordinary concrete, and concrete shear walls made of recycled coarse aggregate were introduced into the steel frame structure. The walls bore the horizontal loads as the main lateral resisting components of the structure when they functioned as the enclosure, thereby providing a new idea for popularizing and applying RAC in steel structure residences. On the other hand, the prefabricated connection between steel frames and concrete shear walls was also considered to facilitate the rapid construction of the project and the timely replacement of the damaged walls. Therefore, the structure of steel frames with infilled RACSWs (SFIRACSWs) is proposed in this paper, and tests on SFIRACSWs were conducted under horizontal low cyclic loading. The effect of two composite forms of steel frames and RACSWs (namely, infilled cast-in-place and infilled prefabricated) on the mechanical behavior of SFIRACSWs is discussed. The failure modes and cooperative mechanism of steel frames and RACSWs is clarified, and the main seismic performance indexes of SFIRACSWs are evaluated comprehensively, providing a theoretical basis for popularizing and applying SFIRACSWs in practical engineering.

## 2. Experimental Program

### 2.1. Specimen Design

To analyze the influence of concrete type, the composite forms of steel frames and RACSWs, and the connecting stiffness of beam–column joints (BCJs) on the hysteretic behavior of SFIRACSWs, this study designed six specimens of one-story and one-bay, at a 1:3 scale, and divided them into three groups. The first and second groups were designed to analyze the effects of the composite forms of steel frames and RACSWs, and the third group was a pure steel frame that was used as a reference specimen.

The first group comprised specimens of infilled cast-in-place RACSWs, which were labeled SPE1, SPE2, and SPE3. Only the wall of SPE1 was made of ordinary concrete, to illustrate the effect of concrete type. The walls of the first group were poured at the construction site and connected with steel frames through shear stubs. The second group consisted of specimens of infilled prefabricated RACSWs,

which were labeled SPE4 and SPE5. The walls were prefabricated in the factory and assembled rapidly with steel frames through T-shape connectors embedded in the walls. Specimen SPE6 belonged to the third group.

The span and height of the specimens were 1050 and 1200 mm, respectively; the beam section was HN $150 \times 100 \times 5 \times 8$, and the column section was HW $150 \times 150 \times 7 \times 10$. Two forms of BCJs, namely, welded–bolted rigid joint and flush end-plate semi-rigid joint, were adopted, as illustrated in Figure 1a,b, to investigate the influence of the connecting stiffness of BCJs on the seismic performance of the SFIRACSWs. Specimens SPE1, SPE2, SPE4, and SPE6 adopted rigid joints, while the other specimens used semi-rigid joints. The main parameters of the specimens are presented in Table 1, and the dimensions and details of the specimens are illustrated in Figure 1.

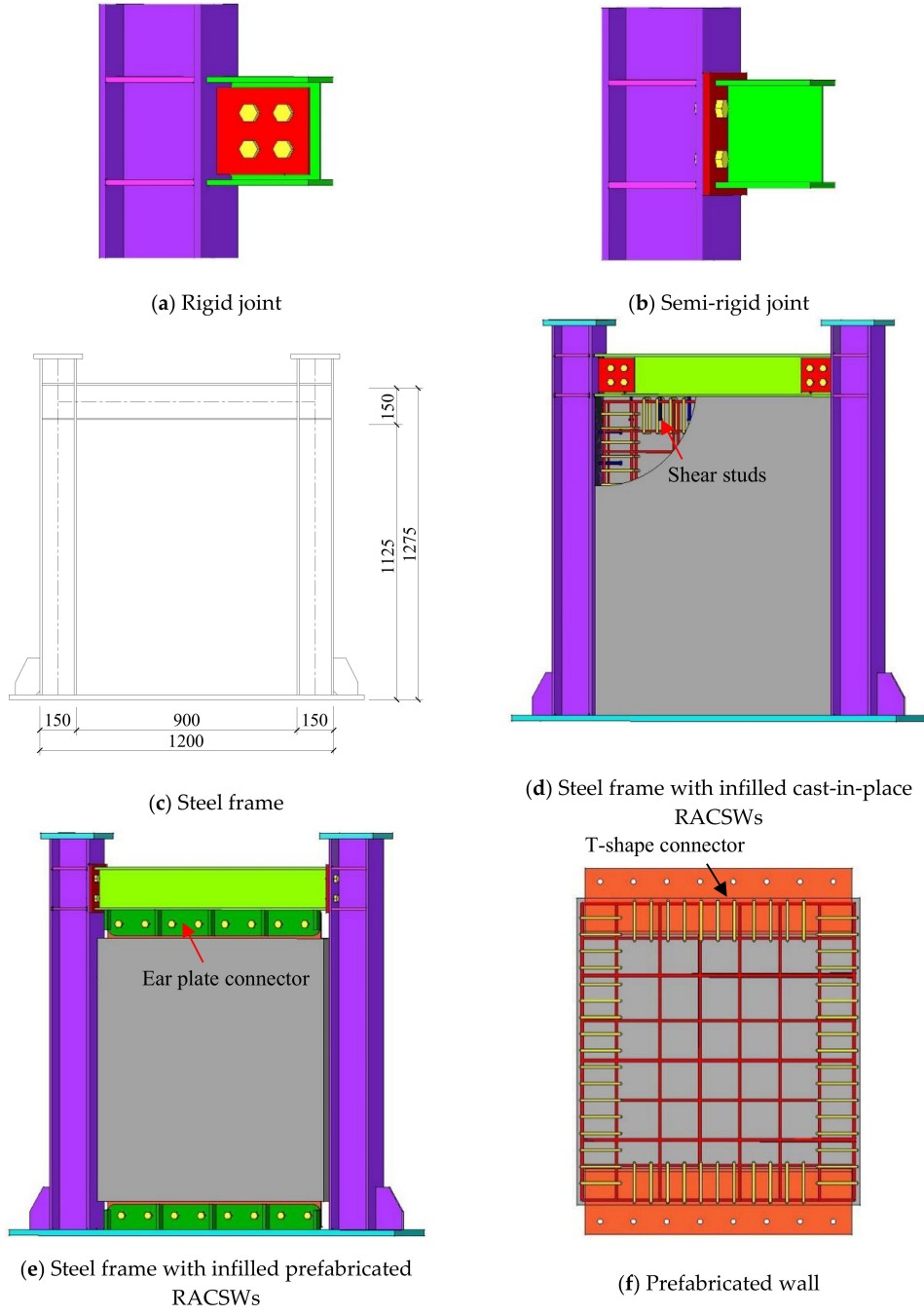

(**a**) Rigid joint

(**b**) Semi-rigid joint

(**c**) Steel frame

(**d**) Steel frame with infilled cast-in-place RACSWs

(**e**) Steel frame with infilled prefabricated RACSWs

(**f**) Prefabricated wall

**Figure 1.** Dimensions and details of specimens.

**Table 1.** Main parameters of specimens.

| Group | Specimen | Beam–Column Joints | Concrete Type | Wall Type | Connection of Walls and Steel Frames | Reinforcement of Walls (mm) |
|---|---|---|---|---|---|---|
| One | SPE1 | Welded-bolted | ordinary concrete | cast-in-place | Infilled shear studs | Double layer, double way Φ6@120 |
| | SPE2 | Welded-bolted | RAC | | | |
| | SPE3 | Flush end-plate | RAC | | | |
| Two | SPE4 | Welded-bolted | RAC | prefabricated | Ear plates, T-shape connectors and bolts | Double layer, double way Φ6@120 |
| | SPE5 | Flush end-plate | RAC | | | |
| Three | SPE6 | Welded-bolted | - | - | - | - |

The diameter of the recycled coarse aggregate was 10–30 mm, and the replacement rate of the coarse aggregate was 100%. The cast-in-place RACSWs with a thickness of 90 mm were connected with steel frames by M16 shear studs, which were welded on the steel frames at 110-mm intervals, as depicted in Figure 1d. The 925 × 860 × 90 mm prefabricated RACSWs were connected with steel frames by ear plates, T-shape connectors, and high-strength bolts. The spacing of the bolts was 100 mm, and the rapid assembly of the walls and steel frames could be achieved as illustrated in Figure 1e. The ear plate connectors were welded on the flange of steel beams and equipped with stiffeners at 200-mm intervals. The T-shape connectors were embedded in the concrete wall and welded with the steel bars of the hidden beams (Figure 1f).

Hidden beams and columns were set around the walls. Four Φ8 steel bars were in the hidden beams and columns, and the diameter of the stirrup spacing of 50 mm was 6 mm. A double-layer steel mesh was arranged on the wall, and the diameter of the horizontal and vertical steel bars with a spacing of 120 mm was 6 mm. The reinforcement details of the infilled walls are presented in Figure 2.

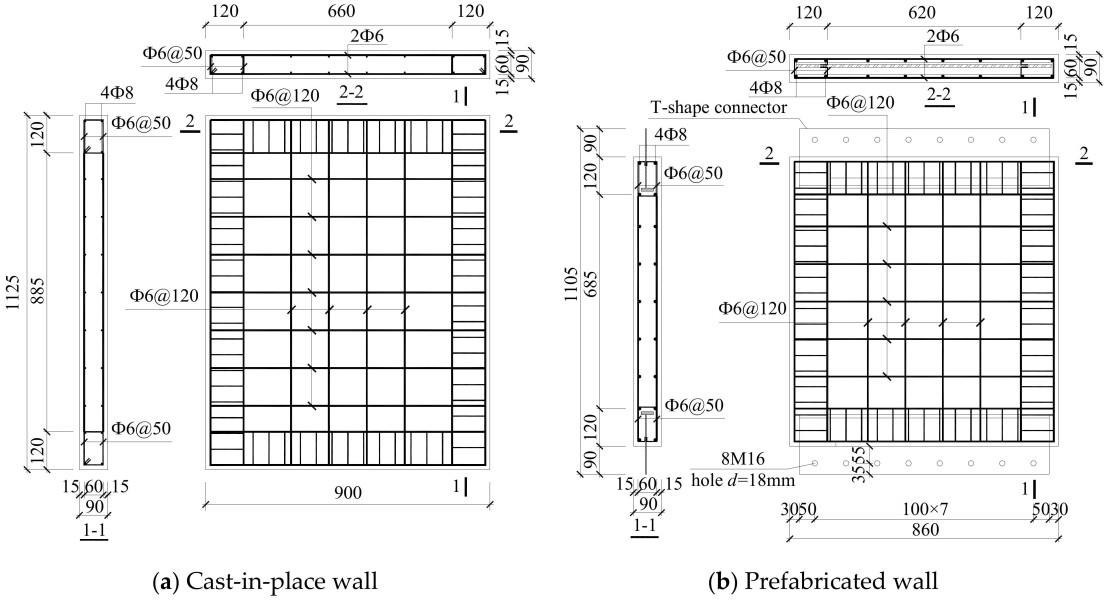

(**a**) Cast-in-place wall          (**b**) Prefabricated wall

**Figure 2.** Reinforcement details of infilled walls.

*2.2. Material Properties*

The tensile strength tests of steel were conducted according to the code *Metallic Materials—Tensile Testing—Part I: Method of Test at Room Temperature*. All steel was Q235B with a yield strength of 235 MPa. The grade of all steel bars was HPB300 with a yield strength of 300 MPa. The mechanical properties of the tested steel are listed in Table 2. The recycled coarse aggregate was from waste concrete specimens that had been placed in the laboratory for many years and were broken into concrete blocks. The design

strength grade of RAC was C30 and the cubic compressive strength was 30 MPa. Test cubes with a size of 100 × 100 × 100 mm were created while pouring walls and cured under the same condition as the walls. The cubic compressive strength of the RAC was measured as 32.8 MPa according to the *Standard for Test Method of Mechanical Properties of Ordinary Concrete*.

**Table 2.** Mechanical properties of steel.

| Interception Position | Thickness (mm) | Yield Stress (N/mm$^2$) | Ultimate Stress (N/mm$^2$) | Young's Modulus (N/mm$^2$) | Elongation at Fracture% |
|---|---|---|---|---|---|
| Beam flange | 8 | 270.20 | 402.30 | $2.09 \times 10^5$ | 31.95 |
| Beam web | 5 | 302.60 | 413.10 | $2.64 \times 10^5$ | 35.15 |
| Column flange | 10 | 268.30 | 447.05 | $2.34 \times 10^5$ | 34.40 |
| Column web | 7 | 283.75 | 452.00 | $2.52 \times 10^5$ | 34.00 |
| Column stiffener | 8 | 281.55 | 403.95 | $1.80 \times 10^5$ | 32.85 |
| Φ6 steel bar | - | 217.30 | 345.50 | $2.50 \times 10^5$ | 32.70 |
| Φ8 steel bar | - | 348.34 | 482.37 | $2.62 \times 10^5$ | 37.65 |

## 2.3. Test Setup and Loading Procedure

The test loading device is illustrated in Figure 3. The specimen was anchored on the ground beam with M30 bolts, and both ends of the ground beam were fixed in the laboratory by pressure beams. The horizontal load was applied by a 1000-kN MTS actuator, and the vertical load was applied by a 1000-kN hydraulic jack. A lateral brace was provided at the end of the MTS actuator, and the outside displacement of the specimen was limited by two groups of pulleys to ensure that the specimen and actuator moved in the horizontal direction. The displacement and strain gauges were arranged in the key parts to study the seismic behavior of the SFIRACSWs, and data were collected by a TDS-630 acquisition instrument. Concrete cracking, buckling of beams and columns, and specimen failure were constantly observed during the test.

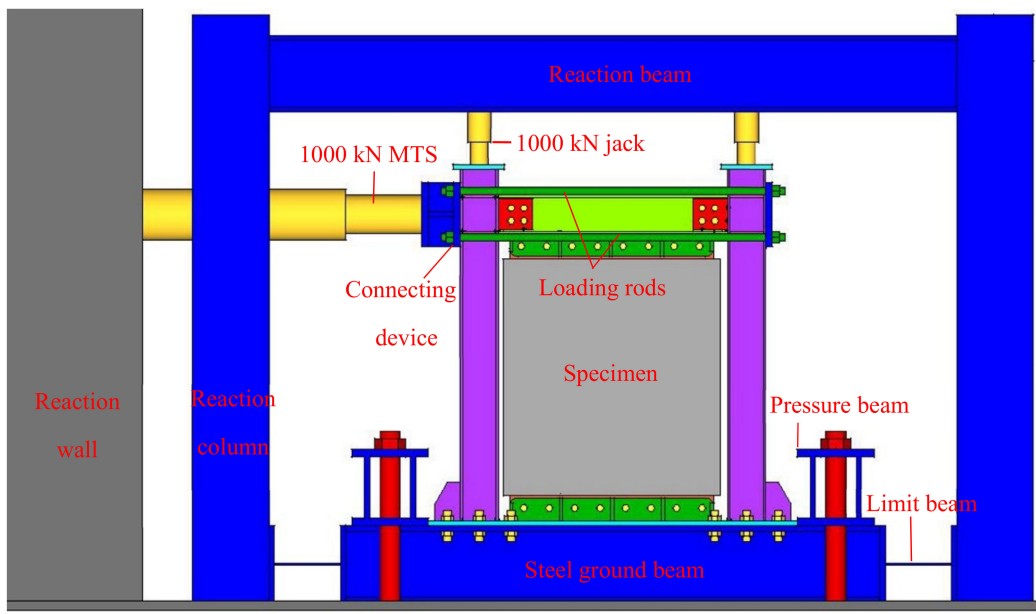

**Figure 3.** Test setup.

In the vertical direction, 250 kN loads were applied to the steel columns, which was calculated by an axial compression ratio of 0.3. The axial compression ratio is the ratio of the design value of the axial load to the product of the total section area and design value of the axial compressive strength of concrete. The horizontal load was applied by the joint control method of force and displacement.

Before the yield of the specimen, the load was controlled by force and cycled once with an increase of 20 kN each time. The load-displacement curves showed a noticeable turning point as a yield sign. After the yield of the specimen, the load was controlled by displacement and cycled thrice with an increase of 0.5 $\delta_y$ each time, which was the estimated yield displacement. Loading stopped when the horizontal load was down to 85% of the peak load.

## 3. Behavior of Test Specimens

### 3.1. General Behavior

#### 3.1.1. Cast-in-Place RACSWs

(1) Specimen SPE1

Specimen SPE1 was in the elastic stage with no observable behavior when the horizontal load was less than 200 kN. Slight oblique cracks began to appear in the middle part of the western side of the wall when the load was 220 kN. Oblique cracks appeared on the lower part of the eastern side of the wall when the load was 260 kN. Then, the specimen yielded locally, and the load was applied by controlling the displacement.

In the displacement control stage, the wall cracks continued to expand, and oblique cracks formed along the 45° direction in the middle of the wall when the displacement was 1.5 $\delta_y$ (Figure 4a). The cracks continued to expand and extend at the control stage of displacement 2.0–4.0 $\delta_y$, and principal cracks with widths of 2–3 mm gradually formed on both sides of the wall (Figure 4b). The concrete along both sides of the principal cracks began to be crushed and fall off, and the width of the cracks reached 5–8 mm (Figure 4c). Local concrete fell off on both sides of the principal diagonal cracks, and the flange at the end of the beam bulged clearly when the displacement was 5.5 $\delta_y$ (Figure 4d). A large area of concrete fell off in the middle of the wall, numerous steel bars were exposed, and the column base buckled outside when the displacement was 6.0 $\delta_y$. At the control stage of displacement 6.5 $\delta_y$, the wall was badly damaged and had holes in the middle. Finally, the horizontal load was reduced by more than 15%, and the specimen lost its carrying capacity.

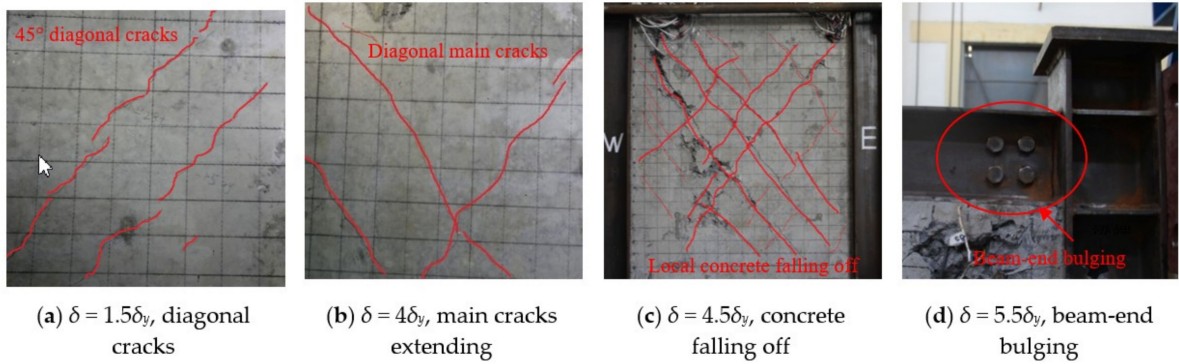

(**a**) $\delta = 1.5\delta_y$, diagonal cracks    (**b**) $\delta = 4\delta_y$, main cracks extending    (**c**) $\delta = 4.5\delta_y$, concrete falling off    (**d**) $\delta = 5.5\delta_y$, beam-end bulging

**Figure 4.** Local failure of SPE1.

(2) Specimen SPE2

Specimen SPE2 was in the elastic stage with no observable behavior when the horizontal load was less than 140 kN. A slight crack appeared on the lower part of the wall when the load was 160 kN. The wall cracks continued to expand when the load was 280 kN. Then, the specimen yielded locally, and the load was applied by controlling the displacement.

In the displacement control stage, several oblique cracks were observed along the 45° direction of the wall, with a length of approximately 100 cm when the displacement was 1.5 $\delta_y$ (Figure 5a). The wall cracks continued to expand and extend to the edge of the wall when the displacement was 2.0 $\delta_y$, thereby forming three principal cracks along the diagonal direction. The local concrete expanded and

fell off at the intersection of the principal diagonal cracks when the displacement was 3.5 $\delta_y$, and the width of the cracks reached 4–5 mm (Figure 5b). A large area of concrete on top of the wall fell off and extended along the principal cracks when the displacement was 4.5 $\delta_y$, thereby forming two dropping areas with a length of approximately 50 cm. The local steel bars were exposed. Numerous steel bars were exposed when the displacement was 5.5 $\delta_y$. Then, the local wall showed holes. The flange at the end of the beam bulged upward (Figure 5c) and the column base buckled (Figure 5d). Finally, the horizontal load was reduced by more than 15%, and the specimen lost its carrying capacity.

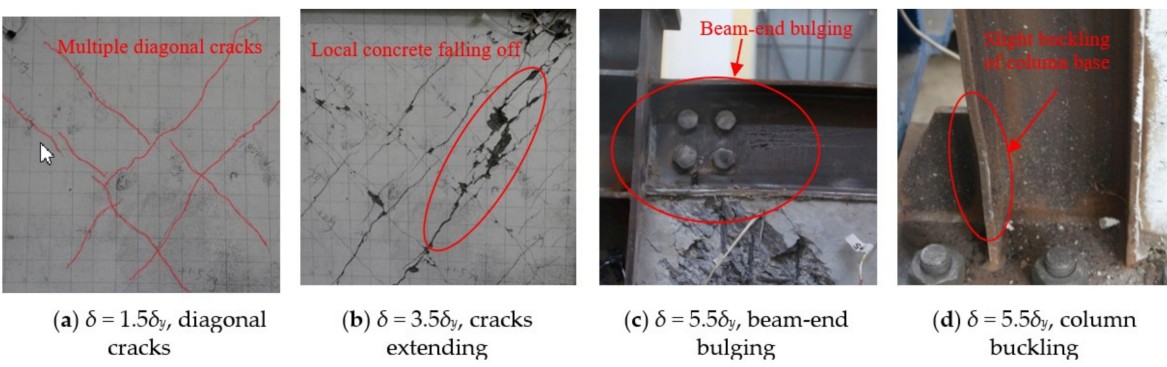

(**a**) $\delta = 1.5\delta_y$, diagonal cracks    (**b**) $\delta = 3.5\delta_y$, cracks extending    (**c**) $\delta = 5.5\delta_y$, beam-end bulging    (**d**) $\delta = 5.5\delta_y$, column buckling

**Figure 5.** Local failure of SPE2.

(3) Specimen SPE3

Specimen SPE3 was in the elastic stage with no observable behavior when the horizontal load was less than 180 kN. Numerous fine cracks appeared on the right side of the wall when the load was 200 kN. The cracks constantly expanded when the load was 280 kN. Then, the specimen yielded locally, and the load was applied by controlling the displacement.

The cracks at the control stage of displacement 1.0–3.0 $\delta_y$ continued to expand and extend to the edge of the wall, thereby forming through cracks with a width of 2–3 mm along the diagonal direction (Figure 6a). The concrete partly fell off on both sides of the principal diagonal cracks when the displacement was 4.0 $\delta_y$, and the width of the cracks reached 4–5 mm (Figure 6b). Considerable concrete fell off at the intersection of the cracks along the diagonal when the displacement was 4.5 $\delta_y$. The local steel bars were exposed. Then, the flange at the end of the beam bulged upward. A large area of concrete fell off when the displacement was 5.5 $\delta_y$. Then, many holes appeared on the wall. Numerous steel bars were exposed and the wall was seriously damaged. Furthermore, the end-plate warped (Figure 6c). The middle part of the column bulged outward and the column base buckled (Figure 6d). Finally, the horizontal load decreased by more than 15%, and the specimen lost its carrying capacity.

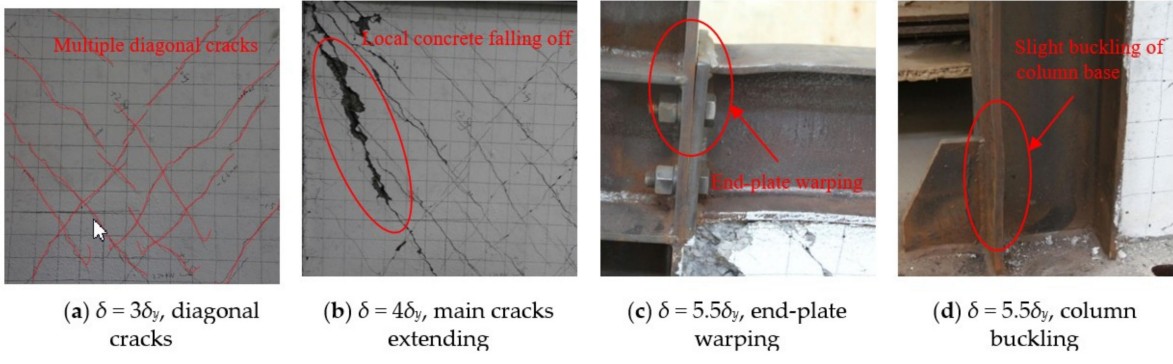

(**a**) $\delta = 3\delta_y$, diagonal cracks    (**b**) $\delta = 4\delta_y$, main cracks extending    (**c**) $\delta = 5.5\delta_y$, end-plate warping    (**d**) $\delta = 5.5\delta_y$, column buckling

**Figure 6.** Local failure of SPE3.

### 3.1.2. Prefabricated RACSWs

(1) Specimen SPE4

Specimen SPE4 was in the elastic stage with no observable behavior when the horizontal load was less than 120 kN. An initial crack appeared at the lower right corner of the wall when the load was 140 kN. Multiple fine oblique cracks appeared in the wall center and extended when the load was 220 kN. The specimen yielded locally, then the load was applied by controlling the displacement.

In the displacement control stage, multiple short cracks with a width of approximately 1 mm appeared along the diagonal direction of the wall when the displacement was 1.5 $\delta_y$. The concrete at the lower right corner of the wall began to fall off, and the number of horizontal cracks gradually increased (Figure 7a). Cracks in the horizontal direction formed at the bottom of the wall when the displacement was 2.5 $\delta_y$ (Figure 7b). The upper ear plate had a relative slip of approximately 10 mm with the T-shape connector (Figure 7c) accompanied by a friction sound among steel plates. Bending deformation occurred on the wall when the displacement was 4.5 $\delta_y$. Then, the corner concrete fell off, exposing the steel bars. The top flange of the steel beam exhibited a brittle fracture when the displacement was 5.0 $\delta_y$ (Figure 7d), and the column base buckled locally. Finally, the horizontal load was reduced by more than 15%, and the specimen lost its carrying capacity.

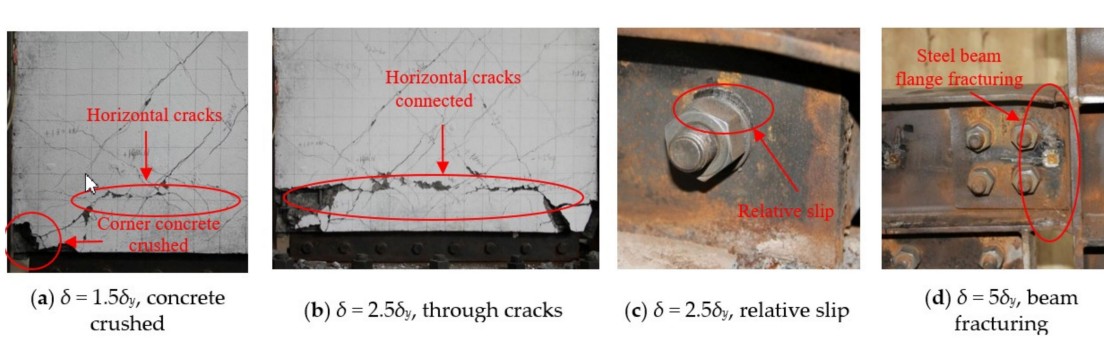

(**a**) $\delta = 1.5\delta_y$, concrete crushed     (**b**) $\delta = 2.5\delta_y$, through cracks     (**c**) $\delta = 2.5\delta_y$, relative slip     (**d**) $\delta = 5\delta_y$, beam fracturing

**Figure 7.** Local failure of SPE4.

(2) Specimen SPE5

Specimen SPE5 was in the elastic stage with no observable behavior when the horizontal load was less than 60 kN. Multiple fine oblique cracks appeared along the diagonal direction of the wall when the load was 80 kN. The number of fine cracks gradually increased and continuously expanded when the load was 120 kN. The specimen yielded locally, then the load was applied by controlling the displacement.

In the displacement control stage, intersecting cracks formed along the diagonal direction of the wall when the displacement was 2.5 $\delta_y$, and the principal cracks were connected (Figure 8a). The concrete at the lower right corner of the wall was crushed and fell off when the displacement was 3.5 $\delta_y$, exposing the steel bars. Multiple horizontal cracks with a width of 2–3 mm appeared at the bottom of the embedded T-shape connector and extended from right to left (Figure 8b). The lower left flange of the steel beam noticeably bulged upward when the displacement was 4.0 $\delta_y$. The upper ear plate had a relative slip of approximately 10 mm with the T-shape connector, accompanied by a friction sound. A large area of concrete fell off when the displacement was 6.0 $\delta_y$. Consequently, the steel bars were exposed, and cracks in the horizontal direction formed at the bottom of the wall (Figure 8c). The end plate warped (Figure 8d), and the column base buckled. Finally, the horizontal load decreased by more than 15%, and the specimen lost its carrying capacity.

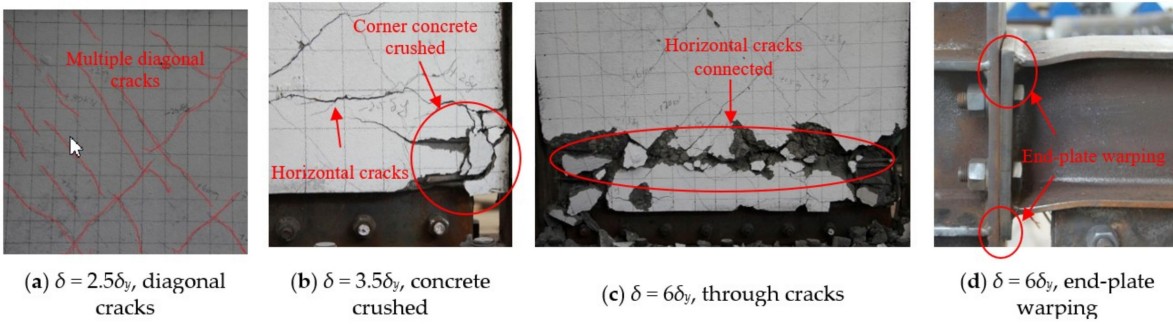

(a) $\delta = 2.5\delta_y$, diagonal cracks   (b) $\delta = 3.5\delta_y$, concrete crushed   (c) $\delta = 6\delta_y$, through cracks   (d) $\delta = 6\delta_y$, end-plate warping

**Figure 8.** Local failure of SPE5.

### 3.1.3. Pure Steel Frame

The specimen SPE6 was in the elastic stage with no observable behavior when the horizontal load was less than 140 kN. The surface coating of the steel column webs fell off locally when the load was 160 kN. A slight bending occurred at the upper flanges of the two columns when the load was 180 kN, and the test entered the displacement control loading stage. The column bases yielded when the displacement was 3.0 $\delta_y$, and a slight out-of-plane instability occurred in the specimen. The flanges at the top portion of the left column and ends of the beam yielded when the displacement was 4.0 $\delta_y$. The structural capacity of the specimen constantly declined and was eventually lost [35].

Several key load points according to the test behaviors of specimens SPE1 to SPE6 are summarized in Table 3.

**Table 3.** Several key load points in the test process.

| Test Process | Specimen SPE1 | Specimen SPE2 | Specimen SPE3 | Specimen SPE4 | Specimen SPE5 | Specimen SPE6 |
|---|---|---|---|---|---|---|
| Elastic stage | Horizontal load $P < 200$ kN | Horizontal load $P < 140$ kN | Horizontal load $P < 180$ kN | Horizontal load $P < 120$ kN | Horizontal load $P < 60$ kN | Horizontal load $P < 140$ kN |
| Cracks appeared | $P = 220$ kN | $P = 160$ kN | $P = 200$ kN | $P = 140$ kN | $P = 80$ kN | - |
| Control loading change point | $P = 260$ kN | $P = 280$ kN | $P = 280$ kN | $P = 220$ kN | $P = 120$ kN | $P = 180$ kN |
| Loading end point | Displacement was 6.5 $\delta_y$ | Displacement was 5.5 $\delta_y$ | Displacement was 5.5 $\delta_y$ | Displacement was 5.0 $\delta_y$ | Displacement was 6.0 $\delta_y$ | Displacement was 4.0 $\delta_y$ |

### 3.2. Failure Modes

From the behavior of the test specimens, the force process of the specimens can be divided into four stages: elastic, concrete cracking, yield, and damage stages. In the elastic stage, steel frames and infilled RACSWs combine to resist exterior loads. The initial stiffness of the structure was high, and the load-displacement curves are linear with no observable behavior.

### 3.2.1. Cast-in-Place RACSWs

The horizontal load applied to the specimens of infilled cast-in-place RACSWs was transferred to the walls by the shear studs. In the concrete cracking and yield stages, initial cracking occurred along the diagonal direction of the walls, and the stiffness of the specimens decreased slightly after the cracking of the walls. The wall cracks gradually expanded and connected with the increase in horizontal load, finally forming three principal cracks with a width of 4–5 mm along the diagonal direction. The local concrete at the intersection of the principal diagonal cracks was crushed and fell off, and the energy was dissipated mainly by the coarse aggregate friction and bite of the cracked surface. A slight bulging deformation emerged at the column base and the end of the steel beam. Upon reaching the peak load, the specimens were in a damaged stage, and a large area of concrete fell

off on both sides of the principal cracks. The steel bars were exposed, and the local wall showed holes. The column base and beam end buckled. The bearing capacity and lateral stiffness of the specimens decreased sharply, and the failure modes are illustrated in Figure 9a.

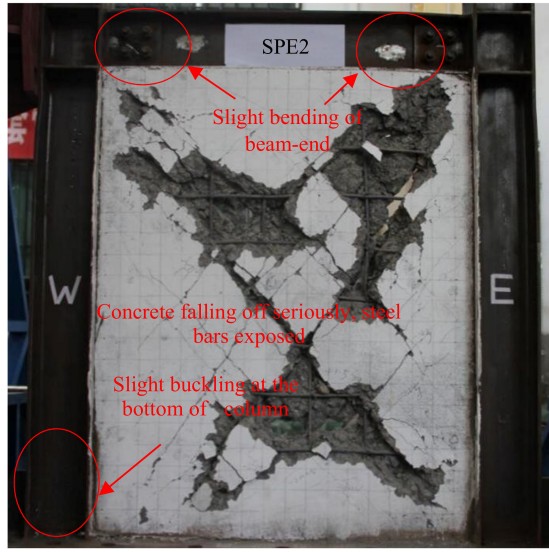 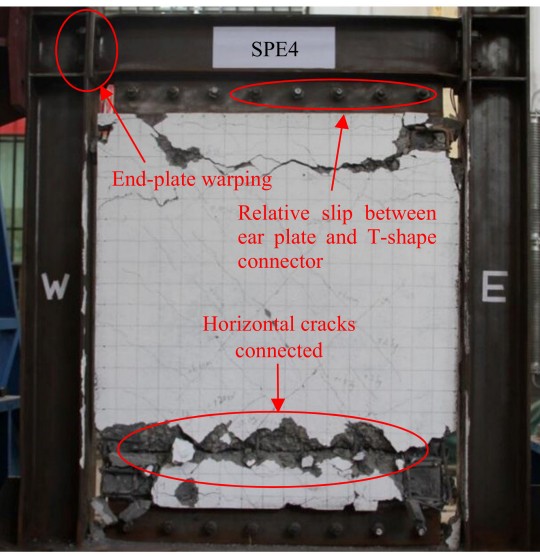

(**a**) Infilled cast-in-place wall        (**b**) Infilled prefabricated wall

**Figure 9.** Failure modes.

### 3.2.2. Prefabricated RACSWs

The horizontal load applied to the specimens of infilled prefabricated RACSWs was transferred to the walls through the ear plates, T-shape connectors, and bolts. In the concrete cracking and yield stages, initial cracking occurred along the diagonal direction of the walls. Horizontal cracks formed at the bottom of the embedded T-shape connectors because of the horizontal shear force. With an increase in horizontal load, the horizontal cracks at the embedded T-shape connectors continued to extend, and principal cracks formed with a width of 3–5 mm. The corner concrete was crushed and began to fall off. The ear plates had a relative slip with the T-shape connectors. After the peak load, a significant amount of concrete fell off, thereby exposing the steel bars. Cracks in the horizontal direction formed at the bottom of the wall. The end plates of the semi-rigid joints warped, and the flange of the steel beam of the rigid joint fractured. The column base buckled. The bearing capacity and lateral stiffness of the specimens degraded rapidly. The failure modes are depicted in Figure 9b.

### 3.3. Transfer Mechanism of Lateral Force

The failure modes of the specimens indicated that the horizontal load of the SFIRACSWs was resisted by the combined steel frames and infilled RACSWs.

### 3.3.1. Cast-in-Place RACSWs

The transfer mechanism of the lateral load of steel frames with infilled cast-in-place RACSWs is illustrated in Figure 10a. In the initial loading stage, the horizontal load was mainly resisted by the compressive strips of the wall along the diagonal direction under the effect of the extrusion pressure of the steel frame and the horizontal shear force transferred by the shear studs. With the increase in horizontal load, the wall was divided into multiple diagonal compressive strips. Then, the concrete at the compressive strips was gradually crushed. The wall gradually failed. The horizontal load was then mostly borne by the steel frame, and the bearing capacity and lateral stiffness of the specimens decreased sharply. The structure of the infilled cast-in-place RACSWs satisfied the requirements of double seismic fortification.

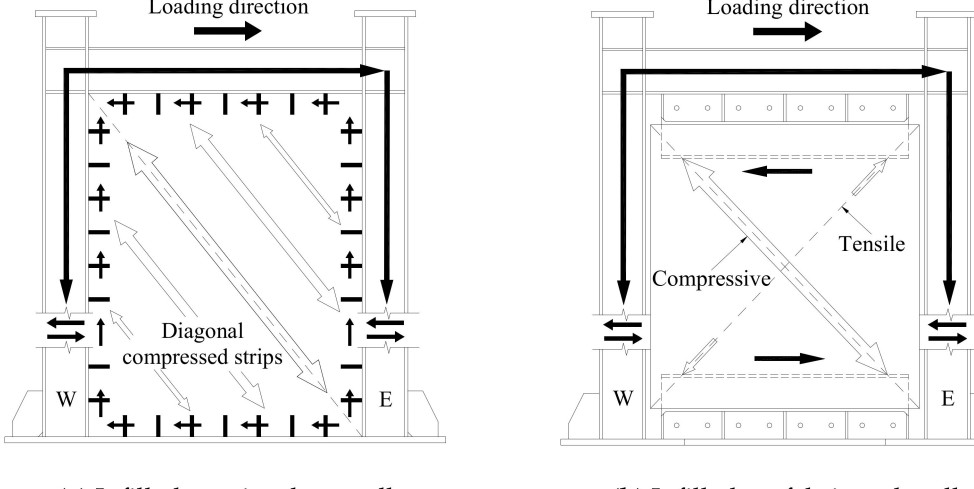

(**a**) Infilled cast-in-place wall　　　　　(**b**) Infilled prefabricated wall

**Figure 10.** Transfer mechanism of the lateral load.

### 3.3.2. Prefabricated RACSWs

The transfer mechanism of the lateral load of steel frames with infilled prefabricated RACSWs is illustrated in Figure 10b. In the early loading stage, the horizontal load was transferred to the wall by the ear plates, T-shape connectors, and bolts, and the wall mainly bore the horizontal shear force. With the increase in horizontal load, the wall also bore oblique compression and tension along the diagonal direction, in addition to the horizontal shear force. The wall began to crack when the stress reached the tensile strength of concrete. In the later loading stage, the wall was divided into multiple diagonal compressive strips. After the cracks in the horizontal direction formed at the bottom of the embedded T-shape connector, a large area of concrete fell off, and the wall gradually failed. The horizontal load was then mainly borne by the steel frame, and the bearing capacity and lateral stiffness of the specimens degraded rapidly.

## 4. Results and Discussions

### 4.1. Hysteretic Curves

The load-displacement hysteretic curves of the specimens are presented in Figure 11.

#### 4.1.1. Cast-in-Place RACSWs

Figure 11a–c demonstrate the following:

(1) The specimens (Figure 11a–c) are in the elastic stage at the initial loading stage, the hysteretic curves are linear, and the loops are narrow. The hysteresis loops become spindle-shaped, and the loops open gradually with the expansion and connection of cracks in the cast-in-place RACSWs. A significant "pinch effect" occurs at the zero point. The hysteretic curves become fully arched and have a reverse S shape after the peak load because of the large area of concrete falling off in diagonal compressive strips and the plastic deformations of the steel beam and columns. The areas enclosed by the loops increase. The bearing capacity of the specimens decreases noticeably under the same load.

(2) Comparison of the hysteretic curves of the SPE1 infilled ordinary concrete wall (Figure 11a) and SPE2 infilled RACSWs with a 100% replacement rate of recycled coarse aggregate (Figure 11b) show that the area and shape surrounded by hysteretic curves and peak loads of the structure are close, thereby indicating that the performance of RACSWs is close to that of the ordinary concrete wall.

(3) The hysteretic curves of specimens SPE2 (Figure 11b) and SPE3 (Figure 11c) are relatively close, thereby showing that the connecting stiffness of BCJs slightly influences the hysteretic behavior of the specimens of infilled cast-in-place RACSWs.

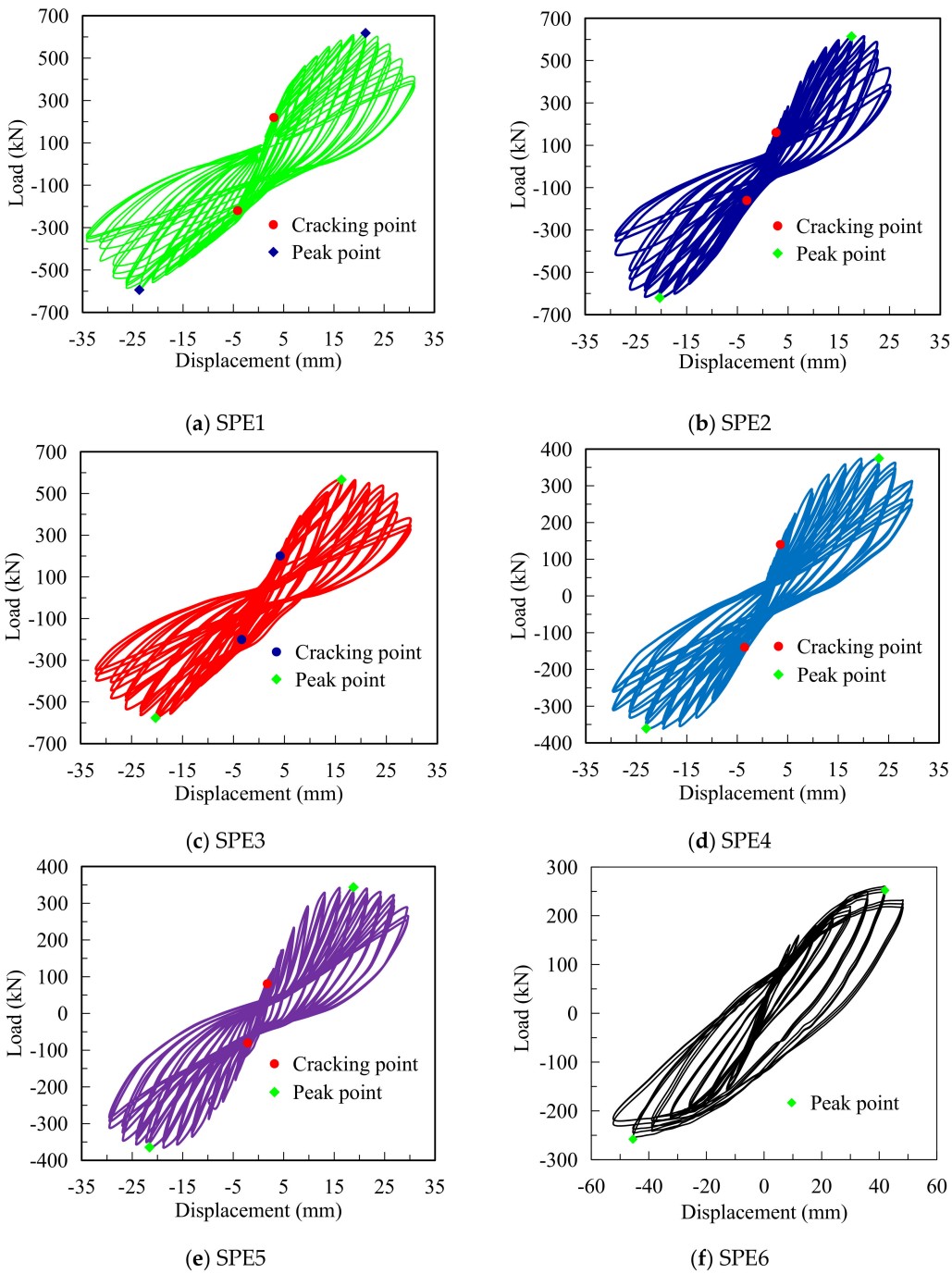

**Figure 11.** Hysteretic curves.

### 4.1.2. Prefabricated RACSWs

Figure 11d,e depict the following:

(1) The stiffness of the specimens (Figure 11d,e) is high at the initial loading stage, and the hysteretic curves are linear. Moreover, no residual deformation occurs after unloading. With the expansion and connection of cracks in the prefabricated RACSWs, the stiffness of the specimens starts to decline and the loops open gradually. Next, a significant "pinch effect" occurs at the zero point. The energy dissipation capacity of the structure is increased, the enclosed areas of the loops increase, and the hysteresis loops are spindle-shaped because of the crushing and collapse of the corner concrete on the wall, the local buckling of the steel frame, and the relative slip between the connectors. Residual deformation occurs after unloading. The hysteretic curves become fully arched after reaching the peak

load. The bearing capacity of the specimens decreases under the same load due to the large relative slip among the connectors.

(2) The hysteresis curves of SPE4 (Figure 11d) and SPE5 (Figure 11e) are nearly coincidental, indicating that the connecting stiffness of BCJs has an insignificant effect on the hysteretic behavior of the specimens of infilled prefabricated RACSWs.

### 4.2. Skeleton Curves

The load-displacement skeleton curves of the specimens are illustrated in Figure 12. The loads of the main characteristic points are summarized in Table 4, where $P_{cr}$, $P_y$, $P_{max}$, and $P_u$ are the cracking, yield, peak, and damage loads of the specimen, respectively, and $P_u = 0.85 P_{max}$. Figure 12 and Table 4 demonstrate the following:

(1) The skeleton curve of the pure steel frame is relatively smooth. The skeleton curves of the specimens are S-shaped when the cast-in-place RACSWs are infilled. Compared with SPE2 infilled RACSWs with a 100% replacement rate of recycled coarse aggregate, the cracking load of SPE1 infilled ordinary concrete wall increases by 37%, the average yield load decreases by 22%, and the bearing capacity is nearly the same.

(2) The bearing capacity of SPE2 is 2.4 times higher than that of the pure steel frame. The load decreases faster in SPE2 and SPE3 than in the pure steel frame after the peak load, demonstrating that the ductility of the specimens of infilled cast-in-place RACSWs decreases slightly.

(3) The comparison of SPE2 and SPE3 demonstrates that the concrete cracking load of the specimen is approximately 1.25 times higher in end-plate joints than in welded–bolted joints. The yield and peak loads decrease by 13% and 8%, respectively, showing that the connecting stiffness of BCJs slightly influences the bearing capacity of the specimens of infilled cast-in-place RACSWs.

(4) The bearing capacity is 1.44 times higher in SPE4 than in the pure steel frame when the prefabricated RACSWs are infilled, thereby indicating that the prefabricated RACSWs can effectively improve the bearing capacity of the structure.

(5) The skeleton curves of SPE4 and SPE5 are coincidental at the initial loading stage, and the peak load is only 4% lower in SPE5 than in SPE4, emphasizing that the connecting stiffness of BCJs slightly influences the bearing capacity of the specimens of infilled prefabricated RACSWs. The load of SPE4 and SPE5 decreases smoothly after the peak load, thereby indicating that the structure of the infilled prefabricated RACSWs has a high safety reserve.

**Table 4.** Loads of main characteristic points on skeleton curves.

| Specimen | Loading Direction | Cracking Point | Yield Point | Peak Point | Failure Point |
|---|---|---|---|---|---|
| | | $P_{cr}$ (kN) | $P_y$ (kN) | $P_{max}$ (kN) | $P_u$ (kN) |
| SPE1 | Positive | 219.49 | 402.20 | 618.14 | 525.42 |
| | Negative | 219.84 | 375.50 | 594.08 | 504.97 |
| SPE2 | Positive | 160.15 | 446.50 | 614.32 | 522.17 |
| | Negative | 160.44 | 500.08 | 620.37 | 527.31 |
| SPE3 | Positive | 200.44 | 437.20 | 566.38 | 481.42 |
| | Negative | 200.56 | 399.01 | 576.57 | 490.08 |
| SPE4 | Positive | 140.12 | 272.60 | 374.62 | 318.43 |
| | Negative | 139.72 | 260.85 | 360.74 | 306.63 |
| SPE5 | Positive | 80.76 | 265.42 | 343.48 | 291.96 |
| | Negative | 80.33 | 262.38 | 364.47 | 309.80 |
| SPE6 | Positive | - | 157.64 | 252.10 | 225.21 |
| | Negative | - | 183.33 | 258.03 | 220.72 |

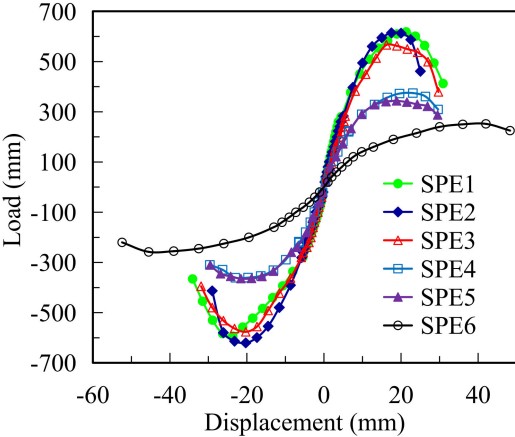

**Figure 12.** Skeleton curves.

### 4.3. Stiffness Degradation

The secant stiffness of the first cycle under the same load is calculated to reflect the degradation law of the stiffness of the specimen under cyclic loading. The formula is

$$K = \frac{|P+| + |P-|}{|\Delta+| + |\Delta-|} \qquad (1)$$

where $P+$ and $P-$ are the positive and negative horizontal loads at the vertex under the same load, respectively; and $\Delta+$ and $\Delta-$ are the corresponding positive and negative horizontal displacements at the vertex under the same load.

The stiffness degradation curves of the specimens are presented in Figure 13. The values of stiffness on the main stages are provided in Table 5, where $\theta$ is the horizontal drift angle of the specimen, and $K_0$ is the initial stiffness of the specimen. Figure 13 and Table 5 present the following:

(1) The stiffness degradation curve of the pure steel frame is relatively smooth. The initial stiffness of SPE2 is 4.3 times higher than that of the pure steel frame when the cast-in-place RACSWs are infilled and approximately 7% lower than that of SPE1 infilled ordinary concrete wall. The comparison of SPE2 and SPE3 implies that the initial stiffness of the specimen is approximately 13% lower in end-plate joints than in welded–bolted joints, and the degradation trend of the stiffness of two specimens is basically the same.

(2) The stiffness of the specimens of infilled cast-in-place RACSWs degrades rapidly at the initial loading stage. With the increase in horizontal load, wall cracks occur and continue to expand. BCJs exhibit a slight rotation, and the stiffness degradation rate of the specimens decreases. The walls are severely damaged and gradually fail after the peak load. The drift angle of the BCJs increases, and the steel frames are used as the second seismic fortification lines to dissipate the seismic energy. The stiffness degradation of the specimens of infilled cast-in-place RACSWs stabilizes.

(3) The initial stiffness is 2.8 times higher in SPE4 than in the pure steel frame when the prefabricated RACSWs were infilled. The stiffness degradation curves of SPE4 and SPE5 are coincidental, indicating that the connecting stiffness of BCJs has an insignificant influence on the stiffness of the specimens of infilled prefabricated RACSWs. The stiffness of the specimens is degraded rapidly at the initial loading stage. The stiffness degradation rate of the specimens decreases, and the stiffness of the specimens of infilled prefabricated RACSWs is steadily reduced by expanding and connecting the cracks in the walls.

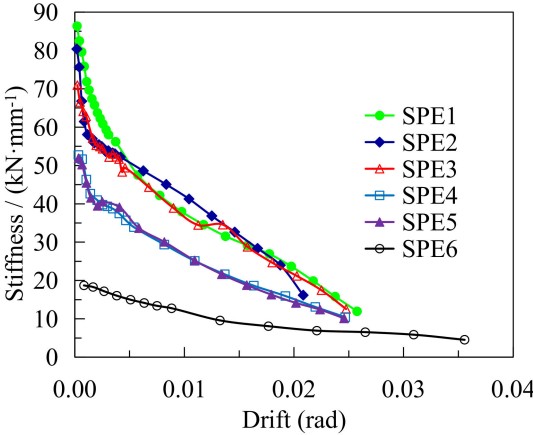

**Figure 13.** Stiffness degradation curves.

**Table 5.** Values of stiffness on main stages.

| Specimen | $K_0$ (kN·mm$^{-1}$) | $\theta$ = 0.001 rad | $\theta$ = 0.005 rad | $\theta$ = 0.010 rad | $\theta$ = 0.015 rad | $\theta$ = 0.020 rad | $\theta$ = 0.025 rad |
|---|---|---|---|---|---|---|---|
| | | $K_1$ (kN·mm$^{-1}$) | $K_2$ (kN·mm$^{-1}$) | $K_3$ (kN·mm$^{-1}$) | $K_4$ (kN·mm$^{-1}$) | $K_5$ (kN·mm$^{-1}$) | $K_6$ (kN·mm$^{-1}$) |
| SPE1 | 86.36 | 73.38 | 50.73 | 37.53 | 30.07 | 23.21 | 13.42 |
| SPE2 | 80.39 | 59.91 | 50.77 | 42.02 | 31.81 | 19.34 | - |
| SPE3 | 70.93 | 62.81 | 48.41 | 36.94 | 30.73 | 21.59 | 12.57 |
| SPE4 | 52.75 | 47.01 | 34.87 | 26.57 | 20.15 | 15.14 | 10.45 |
| SPE5 | 51.89 | 46.21 | 36.27 | 26.86 | 19.64 | 14.32 | 10.12 |
| SPE6 | 18.72 | 18.65 | 15.07 | 11.93 | 8.99 | 7.47 | 6.66 |

## 4.4. Strength Degradation

The degradation coefficient ($\eta$) of the bearing capacity of the same displacement cycle is the ratio of the maximum loads of the last and first cycles. The degradation curves of the bearing capacity of the specimens are depicted in Figure 14. The values of $\eta$ on the main stages are presented in Table 6. Figure 14 and Table 6 demonstrate the following:

(1) The steel frame is a typical flexible structure with good deformation capacity, and its strength degradation is unclear before the peak load. In SFIRACSWs, the walls act as the first seismic fortification lines that resist most of the horizontal load. The concrete at the diagonal compressive strips is gradually crushed and dropped, and the bearing capacity of the structure decreases sharply through the continuous expansion of the cracks in the wall.

(2) The degradation coefficients of the bearing capacity of SPE1, SPE2, and SPE3 are more than 0.97 when the horizontal drift angle is less than 0.01 rad. The degradation coefficients remain more than 0.80 when the horizontal drift angle is 0.02 rad, indicating that the specimens of infilled cast-in-place RACSWs also have a high safety reserve.

(3) The degradation coefficients of the bearing capacity of SPE4 and SPE5 are close when the horizontal drift angle is less than 0.02 rad. The degradation law is consistent, denoting that the damage degree of the specimens of infilled prefabricated RACSWs is the same under the same drift angle. Furthermore, the connecting stiffness of BCJs slightly influences the strength degradation of the specimens of infilled prefabricated RACSWs. The degradation coefficients of the bearing capacity of the specimens of infilled prefabricated RACSWs are more than 0.85.

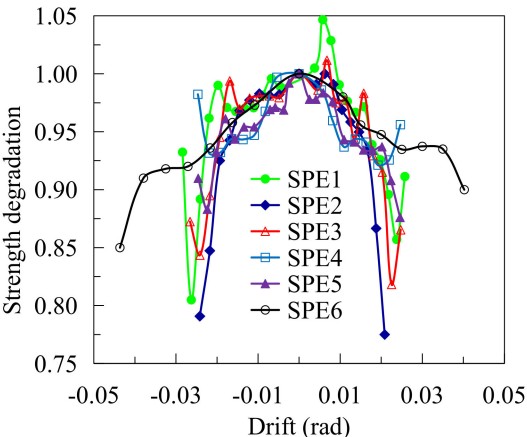

**Figure 14.** Strength degradation curves.

**Table 6.** Values of $\eta$ on main stages.

| Specimen | $\eta$ | | | | | | | | | |
| --- | --- | --- | --- | --- | --- | --- | --- | --- | --- | --- |
| | $\theta =$ −0.025 rad | $\theta =$ −0.020 rad | $\theta =$ −0.015 rad | $\theta =$ −0.010 rad | $\theta =$ −0.005 rad | $\theta =$ 0.005 rad | $\theta =$ 0.010 rad | $\theta =$ 0.015 rad | $\theta =$ 0.020 rad | $\theta =$ 0.025 rad |
| SPE1 | 0.855 | 0.987 | 0.968 | 0.976 | 0.990 | 1.031 | 0.989 | 0.970 | 0.922 | 0.891 |
| SPE2 | 0.781 | 0.904 | 0.963 | 0.982 | 0.982 | 0.994 | 0.973 | 0.947 | 0.812 | - |
| SPE3 | 0.853 | 0.931 | 0.975 | 0.980 | 0.979 | 0.991 | 0.976 | 0.970 | 0.917 | 0.865 |
| SPE4 | 0.982 | 0.932 | 0.944 | 0.954 | 0.997 | 0.987 | 0.945 | 0.942 | 0.922 | 0.956 |
| SPE5 | 0.910 | 0.935 | 0.947 | 0.958 | 0.970 | 0.981 | 0.954 | 0.936 | 0.937 | 0.876 |
| SPE6 | 0.926 | 0.943 | 0.961 | 0.975 | 0.988 | 0.991 | 0.981 | 0.956 | 0.948 | 0.935 |

### *4.5. Ductility Analysis*

Displacement ductility factor is the ratio of damage displacement $\Delta_u$ to yield displacement $\Delta_y$, which is an important index for measuring the deformation capability of a structure. The inter-story drift angles and displacement ductility factors of the main stages are listed in Table 7, where $\Delta_{cr}$, $\Delta_y$, $\Delta_{max}$, and $\Delta_u$ are the cracking, yield, peak, and damage displacements of the specimen, respectively; and $\theta_{cr}$, $\theta_y$, $\theta_{max}$, and $\theta_u$ are the cracking, yield, peak, and damage drift angles, respectively. Table 7 shows the following:

(1) The displacement ductility factor of the pure steel frame is 3.47. The displacement ductility factors of the specimens are from 2.44 to 2.69 when the cast-in-place RACSWs are infilled. The infilled walls can increase the bearing capacity and initial stiffness of the structure while reducing the yield and damage displacement of the structure. Thus, the ductility of the specimens of infilled cast-in-place RACSWs is reduced.

(2) The displacement ductility coefficient in an SPE1 infilled ordinary concrete wall is 1.34 times that of SPE2, indicating that the wall made of recycled coarse aggregate has poor bonding performance and ductility.

(3) The inter-story drift angles are from 1/415 to 1/317 at the concrete cracking stage, from 1/116 to 1/114 at the yield stage, and from 1/66 to 1/64 at the peak point, thereby indicating that the specimens of infilled cast-in-place RACSWs have a good deformation capacity. The displacement ductility factor is approximately 10% higher in SPE3 than in SPE2, suggesting that the specimen of end-plate joints is simple to construct and has a good deformation capacity, and the ductility is better in the end-plate joints than in the welded–bolted joints.

**Table 7.** Displacement and displacement ductility factors.

| Specimen | Loading Direction | Cracking Point | | | Yield Point | | | Peak Point | | | Failure Point | | | Ductility Factors | |
|---|---|---|---|---|---|---|---|---|---|---|---|---|---|---|---|
| | | $\Delta_{cr}$ (mm) | $\theta_{cr}$ | Average | $\Delta_y$ (mm) | $\theta_y$ | Average | $\Delta_{max}$ (mm) | $\theta_{max}$ | Average | $\Delta_u$ (mm) | $\theta_u$ | Average | $\mu$ | Average |
| SPE1 | Positive | 3.07 | 1/391 | 1/332 | 7.73 | 1/155 | 1/136 | 21.30 | 1/56 | 1/53 | 27.43 | 1/44 | 1/42 | 3.55 | 3.28 |
| | Negative | 4.15 | 1/289 | | 9.90 | 1/121 | | 23.71 | 1/51 | | 29.77 | 1/40 | | 3.01 | |
| SPE2 | Positive | 2.64 | 1/454 | 1/415 | 8.78 | 1/137 | 1/114 | 17.51 | 1/69 | 1/64 | 23.80 | 1/50 | 1/47 | 2.71 | 2.44 |
| | Negative | 3.14 | 1/382 | | 12.41 | 1/97 | | 20.31 | 1/59 | | 27.02 | 1/44 | | 2.18 | |
| SPE3 | Positive | 4.15 | 1/289 | 1/317 | 10.30 | 1/117 | 1/116 | 16.20 | 1/74 | 1/66 | 27.41 | 1/44 | 1/43 | 2.66 | 2.69 |
| | Negative | 3.41 | 1/352 | | 10.48 | 1/115 | | 20.30 | 1/59 | | 28.42 | 1/42 | | 2.71 | |
| SPE4 | Positive | 3.55 | 1/338 | 1/337 | 8.92 | 1/135 | 1/137 | 23.06 | 1/52 | 1/52 | 29.03 | 1/41 | 1/41 | 3.25 | 3.32 |
| | Negative | 3.56 | 1/337 | | 8.59 | 1/140 | | 23.02 | 1/52 | | 29.11 | 1/41 | | 3.39 | |
| SPE5 | Positive | 1.76 | 1/682 | 1/619 | 8.59 | 1/140 | 1/139 | 18.83 | 1/64 | 1/60 | 29.13 | 1/41 | 1/41 | 3.39 | 3.40 |
| | Negative | 2.12 | 1/566 | | 8.68 | 1/138 | | 21.56 | 1/56 | | 29.50 | 1/41 | | 3.40 | |
| SPE6 | Positive | - | - | - | 14.15 | 1/85 | 1/83 | 42.00 | 1/29 | 1/27 | 48.28 | 1/25 | 1/24 | 3.41 | 3.47 |
| | Negative | - | - | | 14.88 | 1/81 | | 45.51 | 1/26 | | 52.41 | 1/23 | | 3.52 | |

(4) The displacement ductility factors of the specimens are from 3.32 to 3.40 when the prefabricated RACSWs are infilled. The overall deformation is restrained by the walls, and the horizontal displacements are smaller in the main stages than those in the pure steel frame, although the bearing capacity and lateral stiffness of the specimens of infilled prefabricated RACSWs are remarkably increased. Consequently, the displacement ductility factors are slightly lower in the specimens of infilled prefabricated RACSWs than those in the pure steel frame. The inter-story drift angle is from 1/619 to 1/337 at the concrete cracking stage, from 1/139 to 1/137 at the yield stage, and from 1/60 to 1/52 at the peak point.

### 4.6. Energy Dissipation Capacity

The energy dissipation capacity of the specimens is expressed by the relation curves between the hysteretic loop area and the horizontal drift angle, as shown in Figure 15. The values of energy dissipation in the main stages are presented in Table 8.

(1) The energy of the specimens of infilled cast-in-place RACSWs are dissipated mainly by the flexible deformation of the steel frames and the coarse aggregate friction and bite of the cracked surface of the RACSWs. The energy dissipation is 3.25 times higher in SPE2 infilled cast-in-place RACSWs than in the pure steel frame when $\theta = 0.005$ rad, and 2.6 times higher in SPE2 than in the pure steel frame when $\theta = 0.02$ rad.

(2) The energy dissipation of SPE2 infilled RACSWs is 41% that of SPE1 infilled ordinary concrete wall when $\theta = 0.005$ rad, and 56% that of SPE1 when $\theta = 0.02$ rad. The energy dissipation is approximately 13% higher in SPE3 than in SPE2 when $\theta = 0.005$ rad, and approximately 28% higher in SPE3 than in SPE2 when $\theta = 0.02$ rad, thereby indicating that the end-plate joints are fully deformed and characterized by excellent energy dissipation during loading.

(3) The energy of the specimens of infilled prefabricated RACSWs is dissipated mainly by the flexible deformation of the steel frames and the coarse aggregate friction and bite of the wall cracks and friction slip among the connectors. In the early stage of loading, the cracks on the wall of SPE5 occur early and the concrete cracking load is low; the energy dissipation is slightly higher in SPE5 than in SPE4. With the increase in displacement, the energy dissipation of the two specimens becomes the same. The energy dissipation capacity is approximately two times higher in the specimens of infilled prefabricated RACSWs than in the pure steel frame.

(4) Compared with the pure steel frame, infilled RACSWs can greatly improve the stiffness and energy dissipation capacity of SFIRACSWs while reducing the ductility of the structure, thereby indicating that infilled RACSWs strongly influence the hysteretic behavior of SFIRACSWs.

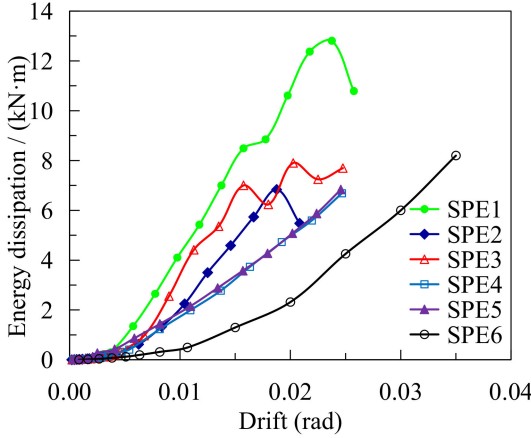

**Figure 15.** Energy dissipation curves.

**Table 8.** Values of energy dissipation at main stages.

| Specimen | Energy Dissipation (kN·m) | | | | |
|---|---|---|---|---|---|
| | $\theta = 0.005$ rad | $\theta = 0.010$ rad | $\theta = 0.015$ rad | $\theta = 0.020$ rad | $\theta = 0.025$ rad |
| SPE1 | 0.96 | 4.27 | 7.93 | 10.83 | 11.55 |
| SPE2 | 0.39 | 2.07 | 4.81 | 6.03 | - |
| SPE3 | 0.44 | 3.38 | 6.45 | 7.71 | 7.70 |
| SPE4 | 0.36 | 1.74 | 3.26 | 4.99 | 6.80 |
| SPE5 | 0.64 | 1.9 | 3.36 | 5.01 | 7.01 |
| SPE6 | 0.12 | 0.45 | 1.30 | 2.32 | 4.25 |

## 5. Conclusions

The following conclusions can be drawn from the low cyclic experiments on steel frames with infilled cast-in-place RACSWs and prefabricated RACSWs:

(1) The bearing capacity and initial stiffness were 2.4 and 4.3 times higher in the steel frames with infilled cast-in-place RACSWs than those in the pure steel frame. The displacement ductility factors were from 2.44 to 2.69. The degradation coefficients of the bearing capacity remained over 0.80 when the horizontal drift angle was 0.02 rad, thereby indicating that the specimens of infilled cast-in-place RACSWs had a high safety reserve.

(2) Compared with the cracking load of the specimen of infilled RACSWs with a 100% replacement rate of recycled coarse aggregate, that of the infilled ordinary concrete wall increased by 37%, the yield load decreased by 22%, and the bearing capacity was nearly the same. These results indicate that the performance of RACSWs was nearly the same as that of an ordinary concrete wall in the structure of steel frames with infilled shear walls.

(3) The yield and peak loads of the specimen decreased by only 13% and 8%, respectively, in the end-plate joints compared with those in the welded–bolted joints. Furthermore, the initial stiffness was reduced by approximately 13%. The infilled cast-in-place RACSWs relieved the rotation deformation of semi-rigid joints and weakened the influence of the connecting stiffness of BCJs on the bearing capacity of the structure of infilled cast-in-place RACSWs.

(4) The bearing capacity and initial stiffness were 1.44 and 2.8 times higher in the steel frames with infilled prefabricated RACSWs than those in the pure steel frame, and the displacement ductility factors were from 3.32 to 3.40. The difference in bearing capacity of the specimens in the welded–bolted and end-plate joints was only 4%, and the turning capability and ductility were better in the semi-rigid joints than in the rigid joints.

(5) The connectors between the steel frames and prefabricated RACSWs were undamaged during the test, and the shear force was transferred successfully. The cracks in the horizontal direction were formed at the connection between the embedded T-shape connectors and the walls, and shear failure occurred in the specimens. Therefore, the connection construction between the embedded T-shape connectors and walls should be given sufficient attention.

(6) The prefabricated shear walls made of recycled coarse aggregate improved the lateral stiffness and bearing capacity of the structure of infilled prefabricated RACSWs. The structure of infilled prefabricated RACSWs was characterized by a favorable deformation capability to satisfy the design requirements of structure behavior in the seismic fortification area. The walls and steel frames could be rapidly installed in the construction field. Furthermore, the structure of the infilled prefabricated RACSWs was safe, highly efficient, convenient to repair and replace after an earthquake, and had a satisfactory engineering application value.

**Author Contributions:** Conceptualization, L.S. and H.G.; methodology, Y.L.; software, Y.L.; validation, L.S., H.G.; and Y.L.; formal analysis, L.S.; investigation, L.S.and H.G.; resources, H.G.; data curation, L.S.; writing—original draft preparation, L.S.; writing—review and editing, L.S. and H.G.; visualization, L.S.; supervision, Y.L.; project administration, H.G.; funding acquisition, H.G. and Y.L.

**Funding:** This research was funded by the National Natural Science Foundation of China No. 51308454.

**Conflicts of Interest:** The authors declare no conflicts of interest.

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
