# Peer review of "Experimental Study on Seismic Behavior of Steel Frames with Infilled Recycled Aggregate Concrete Shear Walls"

_applsci, doi:10.3390/app9214723_

Round 1

Reviewer 1 Report

This paper is well written and the Einglish quality is good.

The reviewer asks though that the authors broaden the introduction to explain alternatives to using concrete shear walls. E.G. steel sheeted walls (like reference 29). This can just be a few sentences but it helps the reader to put the work in context. This should be in the first paragraphs as it helps to introduce the problem.

The literature review does not seem focussed in terms of identifying exactly what research gap is to be filled, and how the results then fit into the literature i.e. express the novelty more clearly. But other than that the quality of the work is good.

This paper may be of interest to the authors (no need to cite).
Bahrebar (2016). Structural performance assessment of trapezoidally-corrugated and centrally-perforated steel plate shear walls. Journal of Constructional Steel Research, 122, 584-594. 10.1016/j.jcsr.2016.03.030

Author Response

This paper is well written and the Einglish quality is good.

Point 1: The reviewer asks though that the authors broaden the introduction to explain alternatives to using concrete shear walls. E.G. steel sheeted walls (like reference 29). This can just be a few sentences but it helps the reader to put the work in context. This should be in the first paragraphs as it helps to introduce the problem.

Response 1: In the structure of steel frames with infilled walls, the stress characteristics of steel plates and concrete plates are different. Steel plates mainly play its tensile capacity, while concrete plates mainly play its compressive capacity. Therefore, the authors think that it is not appropriate to use concrete shear walls as alternatives of steel plate shear walls from the perspective of stress characteristics. And the RAC shear walls used in this paper is reasonable as an alternative to the ordinary concrete shear walls or steel plate shear walls covered with concrete slabs, and it was supplemented in lines 114-116 of the original manuscript.

Point 2: The literature review does not seem focussed in terms of identifying exactly what research gap is to be filled, and how the results then fit into the literature i.e. express the novelty more clearly. But other than that the quality of the work is good.

Response 2: Thank you for your advice. The authors make a supplement to the starting point and innovation of this study. See lines 120-123 of the original manuscript for details.

Point 3: This paper may be of interest to the authors (no need to cite).

Bahrebar (2016). Structural performance assessment of trapezoidally-corrugated and centrally-perforated steel plate shear walls. Journal of Constructional Steel Research, 122, 584-594. 10.1016/j.jcsr.2016.03.030

Response 3: Thank you for your advice. The authors had read the paper mentioned carefully and obtained some help in revising the original manuscript.

Reviewer 2 Report

The paper "Experimental study on seismic behaviour of steel frames wiht infilled recycled aggregate concrete shear walls" is interesting and within the scope of APPlied Sciences. Strategies to reuse the materials, minimize the waste of natural aggregates, and reduce the consuption of natural resources and energy is a very interesting topic from the point of view of environmental sustainability and circular economy.

I have this commenst for the authors:

I would recommend revising the abstract to include in the numerical results of the trials, in order to highlight much more the obtained results.

It would be interesting to indicate the origin of the materials used as recycled aggregates.

Lines 164-166: Indicate why you use this specimen size. What is the standard you use?. Describe the condictions of curing of specimens

Line 279 does no clarify the test behaviour of specimen SPE6.

After line 279, and to better understand the behaviour test for all the specimes will be interesting show the results for all the specimens together in a table. Therefore, you could better compare the results between them

Author Response

The paper "Experimental study on seismic behaviour of steel frames wiht infilled recycled aggregate concrete shear walls" is interesting and within the scope of APPlied Sciences. Strategies to reuse the materials, minimize the waste of natural aggregates, and reduce the consuption of natural resources and energy is a very interesting topic from the point of view of environmental sustainability and circular economy.

I have this commenst for the authors:

Point 1: I would recommend revising the abstract to include in the numerical results of the trials, in order to highlight much more the obtained results.

Response 1: The abstract has been revised and some test numerical results have been added, please see the manuscript for details.

Point 2: It would be interesting to indicate the origin of the materials used as recycled aggregates.

Response 2: The recycled coarse aggregate was from waste concrete specimens, which had been placed in the laboratory for many years and were broken into concrete blocks. Corresponding revision has been made in the manuscript.

Point 3: Lines 164-166: Indicate why you use this specimen size. What is the standard you use? Describe the condictions of curing of specimens.

Response 3: The basis behind the choice in the tested specimens mainly based on two points, the important one is to refer to the specific test specimen size in similar experimental literature, and the other is to estimate the span and height of the specimens, the size of the beam and column and the thickness of the wall according to the size in the actual engineering and the scale ratio.

Point 4: Line 279 does no clarify the test behaviour of specimen SPE6.

Response 4:  The test behaviour of specimen SPE6 has been added to the manuscript.

Point 5: After line 279, and to better understand the behaviour test for all the specimes will be interesting show the results for all the specimens together in a table. Therefore, you could better compare the results between them.

Response 5: Several key load points in the test process are summarized in Table 3, and the specific test behavior is described in the original manuscript.
